# The Polarity of an Amino Acid at Position 1891 of Severe Fever with Thrombocytopenia Syndrome Virus L Protein Is Critical for the Polymerase Activity

**DOI:** 10.3390/v13010033

**Published:** 2020-12-27

**Authors:** Kisho Noda, Yoshimi Tsuda, Fumiya Kozawa, Manabu Igarashi, Kenta Shimizu, Jiro Arikawa, Kumiko Yoshimatsu

**Affiliations:** 1School of Medicine, Hokkaido University, Sapporo 060-8638, Japan; hu-kisho-noda466@eis.hokudai.ac.jp (K.N.); tokagekawaii@eis.hokudai.ac.jp (F.K.); 2Department of Microbiology and Immunology, Faculty of Medicine, Hokkaido University, Sapporo 060-8638, Japan; tsuday@med.hokudai.ac.jp (Y.T.); kshimizu@med.hokudai.ac.jp (K.S.); j_arika@med.hokudai.ac.jp (J.A.); 3Research Center for Zoonosis Control, Hokkaido University, Sapporo 001-0020, Japan; igarashi@czc.hokudai.ac.jp; 4Laboratory of Animal Experimentation, Institute for Genetic Medicine, Hokkaido University, Sapporo 060-0815, Japan; 5Graduate School of Infectious Diseases, Hokkaido University, Sapporo 060-0815, Japan

**Keywords:** SFTSV, L protein, bunyavirus, polymerase activity

## Abstract

Severe fever with thrombocytopenia syndrome virus subclone B7 shows strong plaque formation and cytopathic effect induction compared with other subclones and the parental strain YG1. Compared to YG1 and the other subclones, only B7 possesses a single substitution in the L protein at the amino acid position 1891, in which N is changed to K (N1891K). In this study, we evaluate the effects of this mutation on L protein activity via a cell-based minigenome assay. Substitutions of N with basic amino acids (K or R) enhanced polymerase activity, while substitutions with an acidic amino acid (E) decreased this activity. Mutation to other neutral amino acids showed no significant effect on activity. These results suggest that the characteristic of the amino acid at position 1891 of the L protein are critical for its function, especially with respect to the charge status. Our data indicate that this C-terminal domain of the L protein may be crucial to its functions in genome transcription and viral replication.

## 1. Introduction

Severe fever with thrombocytopenia syndrome virus (SFTSV)—currently also known by the species name Dabie bandavirus—belongs to the genus *Bandavirus*, which is a new member of the family *Phenuiviridae* of the order *Bunyavirales.* This virus is transmitted by ticks and causes severe fever with thrombocytopenia syndrome (SFTS). The mortality rate of SFTS reaches approximately 30% in Japan and is one of the primary public health problems in East Asian countries [1,2,3,4,5,6,7,8,9,10].

The YG1 strain was isolated from the first SFTS patient in Japan [2]. We previously established three subclones from this strain using the limiting dilution method based on the differences in cell fusion activity and reported on their virological characteristics [11]. Subclone B7 shows a strong cytopathic effect (CPE) pattern at pH 7.3 and has two amino acid mutations in the glycoprotein (GP) and one amino acid mutation in the L protein compared to YG1. Subclone A4, which also has the same mutations in its GP but no mutation of the L protein, does not exhibit CPE under the same conditions. In addition, it was previously demonstrated that amino acid substitution in GP is associated with low-pH-dependent fusion activity [11,12,13]. These results indicate that a single amino acid mutation in the C-terminal of the L protein could be responsible for the observed CPE pattern. Whether this mutation causes changes in L protein activity, however, remains unknown.

The L protein is essential for viral genome replication and transcription, not only for bunyaviruses such as SFTSV but also for all segmented negative-stranded RNA viruses. In contrast to studies on the influenza A virus, PA, PB1, and PB2, it remains unclear what functional domains are present in the C-terminal region of the bunyavirus L protein and how they function. Recently, several studies revealed a structural model for the SFTSV L protein, which suggested several domains located at the C-terminal region in addition to the cap-binding domain (CBD) [14,15,16].

In this study, we constructed L protein variants with one amino acid substitution at position 1891 and compared their activities using minigenome and quantitative PCR (qPCR) assays. Position 1891 is suggested to contain one of the functional C-terminal domains of the L protein, and our results demonstrate the possible structural importance of this domain. These results provide the basis for understanding the machinery of the L protein in viral replication in addition to assisting the design of new drugs aimed at targeting the L protein of the bunyavirus.

## 2. Materials and Methods

### 2.1. Cells

BHK/T7-9 cells stably expressing T7 RNA polymerase (kindly provided by Dr. Makoto Sugiyama and Dr. Naoto Ito, Faculty of Applied Biological Science, Gifu University) were maintained in minimum essential medium (Gibco, Thermo Fisher Scientific, Life Technologies, Waltham, MA, USA) supplemented with heat-deactivated 5% fetal bovine serum (Biowest, Nuaille, France), 10% tryptone phosphate broth (Becton, Dickinson and Company, Sparks, MD, USA), and 1% penicillin/streptomycin (Sigma Aldrich CO., St. Louis, MO, USA). Cells were selected by hygromycin B (Invitrogen, Thermo Fisher Scientific, Life Technologies, Waltham, MA, USA) every 10 passages. HEK293T cells were maintained in Dulbecco’s modified Eagle’s medium—high glucose (Sigma Aldrich CO.) supplemented with heat-deactivated 10% fetal bovine serum and 1% penicillin/streptomycin.

### 2.2. Construction and Expression of SFTSV L Protein Variant Plasmids

We used the plasmids pATX-SFTSV LIII and pCSFTSV-L as described by Lundu et al. for the preparation of constructs to express mutant L proteins [17]. Based on the SFTSV YG1 strain L protein open reading frame sequence (GenBank: AB817979.1), we designed primers for the following single amino acid substitutions at position 1891 (Table 1): asparagine (N) to lysine (K), arginine (R), glutamic acid (E), alanine (A), or glutamine (Q). The mutations were introduced into pATX- SFTSV LIII using the inverse PCR method. Using the general PCR methods, whole plasmid of pATX- SFTSV LIII was amplified and linearized with primers containing the mutation at the 5′ prime end of forward primers in 10 cycle running (Table 1). Then, the PCR products were circularized into the pATX-SFTSV LIII variants with T4 DNA ligase. Each pATX-SFTSV LIII variant was digested with the restriction enzymes, KpnI and SpeI, and fragments containing the mutations were ligated into pCSFTSV-L. The L protein variant plasmids pCSFTSV L_N1891 (wild-type; WT), pCSFTSV-L_N1891K, N1891R, N1891E, N1891A, and N1891Q were generated and purified using an endotoxin-free maxiprep kit (QIAGEN N.V., Hilden, Nordrhein-Westfalen, Germany). The sequences of each variant were checked by Sanger sequencing prior to use in protein expression.

### 2.3. Indirect Immunofluorescence Assay

BHK/T7-9 cells were seeded into 12-well plates and transfected with 1.0 μg of each pCSFTSV-L using a TransIT LT1 reagent (Mirus Bio LLC, Madison, WI, USA), according to the manufacturer’s protocol. The cells were incubated at 37 °C for 24 h and re-seeded onto a 24-well slide. Twenty hours after re-seeding, the cells were fixed with acetone. The fixed cells were blocked using 1% bovine serum albumin (BSA) in phosphate-buffered saline (PBS) and stained using an indirect immunofluorescence assay (IFA). L protein variant antigens were then labeled with rabbit anti-SFTSV L polyclonal antibody ([17] and raised against the 15 amino acid synthetic peptide (386–400) region of L) diluted 1:300 in PBS. Alexa Fluor 488 conjugated anti-rabbit antibodies (Thermo Fisher Scientific, Life Technologies, Waltham, MA, USA) diluted 1:5000 in PBS were used as the secondary antibodies. The nuclei were stained with 4′,6-diamidino-2-phenylindole (DAPI). Fluorescence images were acquired using a Nikon ECLIPSE Ti laser fluorescent microscope (Nikon corporation/Nikon Instech Co., Ltd., Minato-city, Tokyo, Japan) with a 10× lens and NIS Elements (Nikon corporation). The exposure time for picture acquisition was 1.5 s for the SFTSV L protein antigen and 0.5 s for the DAPI-stained nuclei.

### 2.4. Western Blot Assay

HEK293T cells were seeded onto 12-well plates and transfected with 1.0 μg of each pCSFTSV-L variant plasmid using TransIT LT1. Then, 48 h after transfection, the cells were rinsed once with PBS, which was followed by lysis in 4× Laemmli sample buffer (Bio-Rad Laboratories, Inc., Hercules, CA, USA) supplemented with 10% 2-mercaptoethanol and incubation for 5 min at 95 °C. The samples were then subjected to Western blot analysis. Sodium dodecyl sulfate-polyacrylamide gel electrophoresis (SDS-PAGE) was performed using 5%–20% gradient gels, which was followed by transfer of the proteins to a 0.45-μm pore immunoblot polyvinylidene fluoride membrane (Millipore, Billerica, MA, USA). Membranes were blocked for 24 h at 4 °C with Blockace (KAC Co., Ltd., Kyoto city, Kyoto, Japan). The membrane was incubated with rabbit anti-L peptide polyclonal antibody diluted 1:1000 with Can Get Signal^®^ Immunoreaction Enhancer Solution 1 (Toyobo Co., Ltd., Osaka city, Osaka, Japan) and horseradish peroxidase-conjugated anti-rabbit IgG goat IgG as the secondary antibody (Jackson Immuno Research Laboratories, Inc., Baltimore, MD, USA), diluted 20,000 times with Can Get Signal^®^ Immunoreaction Enhancer Solution 2 (Toyobo). For standardization, the membrane was incubated with peroxidase-conjugated anti-glyceraldehyde 3-phosphate dehydrogenase (GAPDH) monoclonal antibody (Fujifilm Wako Pure Chemical Corporation, Osaka city, Osaka, Japan), diluted 1:10,000 with Can Get Signal^®^ Immunoreaction Enhancer Solution 2 immediately after blocking. The membranes were washed with 3 × PBS for 5 min after each antibody incubation. After the final wash, the membranes were incubated with enhanced chemiluminescence (ECL) Prime Western Blotting Detection Reagents (GE Healthcare, Chicago, IL, USA) followed by image acquisition with an ImageQuant LAS 4000 mini (GE Healthcare) using optimal settings.

### 2.5. Minigenome Assay

Previous studies suggested that the M segment-based minigenome system has higher reporter activity than other segment-based systems [18,19]. Based on these results, we also constructed an M segment-based SFTSV minigenome system. A pATX-SFTSV M-segment-based minigenome plasmid expressing Renilla luciferase (RLuc) was constructed as a reporter (pMG-RLuc) for the minigenome system. Next, 96-well plates of BHK/T7-9 cells stably expressing T7 RNA polymerase were separately transfected with 0.02 μg of pCSFTSV-L corresponding to each variant plasmid, 0.02 μg of the pCSFTSV-N-expressing N protein [17], 0.02 μg pMG-RLuc, and 0.025 μg of pCAGGS-FLuc expressing firefly luciferase (FLuc) as a standardization control using TransIT LT1. Twenty-four hours post-transfection, the cells were lysed with a passive lysis buffer (Promega, Madison, WI, USA) and assayed for FLuc and RLuc activity using a dual-luciferase reporter assay system (Promega, Madison, WI, USA). Values for RLuc activity divided by FLuc (resulting in standardized relative light units [sRLU]) were used to standardize differences in transfection efficiency and cell density.

To evaluate RNA replication and transcription, 12-well plates of BHK/T7-9 cells were transfected with 0.1 μg of the pCSFTSV-L variant plasmid (L-WT, N1891K or N1891E), 0.1 μg of pCSFTSV-N, 0.1 μg of pMG-RLuc, and 0.01 μg of pCAGGS-FLuc using TransIT LT1. Twenty-four hours post-transfection, the cells were collected, and 10% of the cells were lysed with PLB for a dual-luciferase reporter assay, as described above. The total RNA extracted from the rest of the transfected cells was purified with a RNeasy mini plus kit (QIAGEN) with removal of contaminating DNA performed using a TURBO DNA-free™ Kit (Invitrogen, Thermo Fisher Scientific) for quantitative PCR.

### 2.6. Quantitative PCR (qPCR) Assay

For each of the purified total RNA samples acquired in Section 2.5, 0.5 μg was used as a template for reverse-transcription PCR (RT-PCR). The RT-PCR reaction was performed using ReverTra Ace (Toyobo) with each of the corresponding segment-specific primers (Table 2) or oligo(dT) primer, and 20 μL of the RT-PCR products were diluted with DDW up to 100 μL. The qPCR reaction was performed with a KAPA SYBR^®^ FAST qPCR Master Mix (2×) Kit (Sigma Aldrich CO., St. Louis, MO, USA) using each primer pair (Table 2), and qPCR was performed using a LightCycler 480II (Roche Applied Science, Penzberg, Oberbayern, Germany) with the optimized SYBR program.

### 2.7. In Silico Structural Analysis

The Protein Data Bank (PDB) model of the SFTSV L protein (6L42.pdb) reported by Wang et al. [16], lacking the loop structure at positions 1889 to 1896. Based on the L protein model (6L42.pdb), the L protein structural model with the intact 1889–1896 loop structure was estimated using a Molecular Operating Environment (MOLSIS Inc., Tokyo, Japan). From among the top 1000 candidates, the model with the highest score was used for further analysis. The structural models of L protein variants and their surface charge models were generated using Mutagenesis and Advanced Poisson-Boltzmann Solver (APBS) Electrostatics function of PyMOL (Schrödinger, Inc., New York, NY, USA), respectively.

## 3. Results

### 3.1. Estimating the Location of Position 1891 on the Structural Model of the SFTSV L Protein

Bunyavirus L protein has two main functional domains: RNA-dependent RNA polymerase activity and endonuclease activity. Previous structural studies reported functional domains of the SFTSV L protein [14,15]. Sequence analysis of the SFTSV YG1 subclones revealed an N1891K amino acid mutation in the L protein located in the C-terminal domain (Figure 1A). Recently, Wang et al. reported a PDB model of the whole SFTSV L protein (6L42.pdb) including the C-terminal domain. However, this model lacked the loop structure at position 1889–1896 [16]. Therefore, a model of the whole SFTSV L protein, including amino acid positions 1889–1896, was estimated using MOLSIS, and the 1889–1896 loop-structure-patched model of the SFTSV L protein was used for predicting the position of 1891 on the tertiary structure (Figure 1B). In this model, N1891 is located in the arm domain on the back surface of the L protein and close to the border between the arm and lariat domains.

### 3.2. Minigenome Assay for SFTSV L Protein Variants

The amino acid at position 1891 is located in the C-terminal domain and not within a known functional domain, such as the RNA-dependent RNA polymerase domain and the endonuclease domain. However, this amino acid mutation was found only in subclone B7, which showed unique characteristics such as strong CPE induction and clear plaque formation compared to other subclones, even though no clear difference was observed for in vitro growth kinetics [11], suggesting that an amino acid mutation at position 1891 affects the biological function of the L protein. We first generated an in vitro SFTSV minigenome assay to test the SFTSV L protein variants’ viral genome replication and transcription activity. Using the established minigenome system, we first compared the polymerase activity of L_WT and the L-mutant N1891K. The substitution to lysine from asparagine (N1891K) enhanced RLuc activity by 1.5~1.8-fold compared to L_WT (Figure 2A). Next, we compared the amino acid sequence elements between SFTSV strains and Heartland virus, belonging to the same genus *Bandavirus*, as isolated in the United States of America. The amino acid at position 1891 is highly conserved in SFTSV, and the estimated position for 1891 in the Heartland virus corresponds to the non-polar amino acid serine (Figure 2B). We next constructed SFTSV L protein variants to evaluate the influence of amino acid polarity at position 1891 on polymerase activity. To confirm the protein expression of each variant, BHK/T7-9 cells were transfected with the constructed SFTSV L protein-expressing plasmids and fixed with acetone. Expression of the L protein variants was detected by IFA. There were no significant differences in L protein distribution and expression levels among the variants (Figure 2C). Moreover, HEK293T cells were also transfected with L protein variant plasmids for Western blot assays. The Western blot results also showed that all L protein variants were equally expressed in the transfected cells (Figure 2D). These results indicate that a single amino acid substitution at position 1891 of the SFTSV L protein does not affect its expression level or subcellular localization. The substitution to the basic amino acid arginine (N1891R) enhanced RLuc activity by 1.5~1.8-fold compared to L_WT, which is similar to the results observed with lysine (Figure 2A). By contrast, substitution of the acidic amino acid glutamic acid (N1891E) reduced the activity to 10% that of the wild-type (WT). Substitution to the non-polar amino acid, alanine (N1891A), did not significantly affect RLuc activity, while glutamate substitution (N1891Q) slightly enhanced this activity by around 1.3-fold compared to L_WT. These data indicate that the characteristics of the amino acid at position 1891 are important for the regulation of L protein functions, such as transcription and replication of the viral genome, as substitution with a negatively charged acidic amino acid that significantly disrupts L protein activity. Although N1891E showed highly reduced reporter activity compared to L_WT, this activity was still 400 times higher than that of L− (the negative control lacking L protein expression).

### 3.3. qPCR Assay under the Minigenome System Conditions

The RLuc activity of the minigenome assay reflects both the replication and transcription activities of the L protein variants. We next evaluated the positive and negative-sense RLuc RNA expression of L_WT, 1891K, 1891E, and L− as negative controls under minigenome assay conditions. At the same time, we also measured the amount of mRNA of the SFTSV L protein to confirm the expression of the L protein from the plasmid. The results of the minigenome assay from the 12-well plate was consistent with the previous results (Figure 3A). The levels of mRNA corresponding to the SFTSV L protein were significantly higher than that in L−, but no significant differences were found among the variants (Figure 3B). The same result was found for protein detection by IFA and a Western blotting assay, as shown in Figure 2B,C. The quantities of positive-sense RLuc RNA were significantly higher in L_WT and N1891K expressing cells compared with L−. However, there were no significant differences among the SFTSV L variants (Figure 3C). Similar levels of positive-sense RLuc RNA in N1819E and L− were detected. This result suggests nonspecific pathways replicating positive-sense RNA from the negative-sense RNA, which is independent of viral ribonucleoprotein (RNP) complex activity. Assuming that this positive-sense RNA detected in L− reflects the background, this RNA indicated the same tendency as the minigenome assay. N1891K presented the highest amount of positive-sense Rluc RNA followed by L_WT, while N1891E presented the lowest. The amounts of negative-sense RLuc RNA showed no significant differences among the variants or for L−, which was mostly transcribed from the plasmid via T7 polymerase (Figure 3D).

### 3.4. In Silico Surface Charge Analyses of the SFTSV L Protein Variants

Previous results indicated that the electrical charge of the amino acid substituted at the position 1891 is correlating with the L protein activity. We then evaluated the influence of the residue at position 1891 on the surface charge states of the L protein variants based on the generated structural model. The surface charge around position 1891 formed an “edge” on the arm domain side (white dashed circle) that is located at the border between the domains of arm and lariat forms “valley” (yellow dashed line). The charge of the edge remained in a relatively positive charge state toward the border in WT (Figure 4A). As the residue at position 1891 substituted, the charged states of the edge became affected. The edge became especially negatively charged when glutamic acid was substituted, while basic amino acid substitutions enhanced the positive charge of the surroundings (Figure 4B). It was suggested that the lariat domain is also relatively, positively charged.

## 4. Discussion

Like other negative-strand RNA viruses, the L proteins of SFTSV play essential roles in the mediation of replication for both the primed synthesis of mRNA and the non-primed synthesis of complement RNA from the template viral RNA. Therefore, the L protein has to perform multiple functions, for example, as RNA-dependent RNA polymerase, endonuclease, and cap-snatching, which are related to transcription and genome replication [20]. The cap-snatching mechanism has been most thoroughly characterized in the influenza virus, and the cap-binding domain is located in the third subunit, PB2, which is equivalent to the C-terminal region of the Bunyavirus L protein [21,22,23,24,25]. Recent studies revealed the structure of the cap-binding domain of the bunyaviruses including Rift Valley Fever virus (RVFV) and SFTSV, which are located in the C-terminal region of the L protein. This is necessary for cap-snatching alongside the endonuclease domain located at the N-terminal of the L protein [14,15,16,26,27].

In this study, we demonstrated that amino acid substitution in the C-terminal domain at position 1891 influences L protein activity. The minigenome assay results indicated the clear influence on polymerase activity when substituting the residue at position 1891. However, these results did not reveal whether only transcription was affected or if genome replication was also affected. The qPCR assay using minigenome assay conditions showed that the amounts of positive Rluc RNA exhibited the same tendency as those in the minigenome assay, even though the amount of positive Rluc RNA did not show differences as significant as those in the minigenome assay. The effects of this amino acid substitution on viral genome replication were relatively limited when compared to those of transcription, which was shown by the minigenome assay, assuming that the result for the positive Rluc RNA amount reflected the different amounts of mRNA. Our results suggest that the amino acid substitution in the C-terminal domain of the SFTSV L protein, which we found in the subclone, is related to the core function of polymerase activity. Jérôme et al. reported that the C-terminal domains of the RVFV L protein located downstream from the cap-binding domain are also involved in transcription but not replication [14,28]. Thus, the C-terminal domains of the SFTSV L protein also have important roles in facilitating the transcription activity of the SFTSV L protein but only has a limited influence on genome replication, which is consistent with the previous study on RVFV.

Recent studies have also suggested the presence of potential functional domains in the C-terminal region and conformational structures required for viral transcription initiation, in addition to the cap-binding domain [14,15,16]. The results of the minigenome assay are correlated with the electric charge of the of the amino acid sidechain, rather than the size of the sidechain. The positively charged basic amino acids enhanced polymerase activity while this was decreased by negatively charged glutamic acid, suggesting that electric interactions around amino acids are an important factor for effective function. The results of in silico assay suggested that position 1891 is located near the border between the arm and lariat domains and the substitutions to the position influence the charge state of the arm domain near the border. In Wang’s proposed model, separation of the lariat domain from the arm domain triggers viral mRNA transcription [16]. In this case, the surface charge states of each domain might interfere with the flexibility of the lariat domain, where the negatively charged state due to the N1819E substitution causes the arm domain to stick to the lariat domain, resulting in lower transcription efficiency. This result suggests that this charged state difference influences the interactions with factors related to viral replication and highlights the importance of the structure and surface charge of this domain for the function of the SFTSV L protein. Further studies revealing the whole machinery for viral replication of SFTSV are required, including a detailed analysis of viral genome replication. In addition to that, we used the computational algorithm to estimate the structural model with the 1889–1896 loop structure since the reported structural PDB model (6L42.pdb) lacked the 1889–1896 loop. It is presumed that this loop domain is structurally unstable and has a fluctuating structure. However, since this estimated model reflects neither the structural interactions to the residue at position 1891 nor electrical influences brought by other amino acids within the 1889–1896 loop, the results of the in silico assay has a methodological limitation. Further analyses including more detailed computational modeling, or structural analysis are also necessary for understanding the concrete structural mechanism of SFTSV L protein activity.

The N1891K mutant L protein was found in subclone B7 and showed clear cytopathic effects in infected cells compared to those infected with the original YG1 strain or other subclones, which possesses 1891N. Subclone B7 was isolated from the YG1 strain as a diverse quasi-species, and the ratio of B7 to the whole SFTSV population was estimated to be 1/1,000,000 [11,29], which means that subclone B7 is a very minor population. In this study, we demonstrated enhanced polymerase activity resulting from N1891K amino acid substitution. A greater amount of protein aggregation was generally observed in B7-infected cells during IFA experiments when compared with infection with subclone E3 (data not shown). On the other hand, our previous study indicates there was no clear difference in virus titer between B7 and other subclones [11]. This mutation might result in the hyper-synthesis of viral proteins and subsequent host cell stress, which was followed by cell death through enhanced L protein transcription capacity. Furthermore, the natural selection might not work favorably for the subclone B7 in which enhanced L protein capacity and higher cytopathogenicity are trading off each other. Otherwise, the whole SFSTV subclones infecting in the same host cells might be able to share the advantage of enhanced capacity brought by this mutant L protein. The C-terminal region of the SFTSV L protein might play an important role in host cell death as well as pathogenicity. One of the most famous examples of how the C-terminal domains of viral polymerase influence not only viral replication but pathogenicity and host ranges is the PB2 domain of influenza virus A [30,31,32,33]. Both in vitro and in vivo investigations are still needed to reveal the relationship of the SFTSV L protein, including this domain, with host pathogenicity. Since subclone B7 was isolated by limiting the dilution method on a cell culture system and only evaluated the character in in vitro assays, we need to evaluate the virological fitness of the subclones in vivo model. Additionally, studies using deep sequencing of clinical specimens and wild tick samples are also necessary to evaluate the quasi-species of SFTSV and the functional ability of SFTSV in viral replication and pathogenicity. The results of such studies may provide the basis for determining SFTSV’s history, such as changes in its tropism and viral dynamics.

In summary, we demonstrated that the surface charge of the amino acid at position 1891 of the SFTSV L protein has a significant influence on polymerase activity. The in silico analysis suggested the possible structural importance of this position, including its surface charge and interactions with other surrounding domains or host factors. These findings provide the basis for understanding the role of the C-terminal domains of the L protein in the viral replication of SFTSV and its relationships with pathogenicity, viral replication, and evolution.

## Figures and Tables

**Figure 1 viruses-13-00033-f001:**
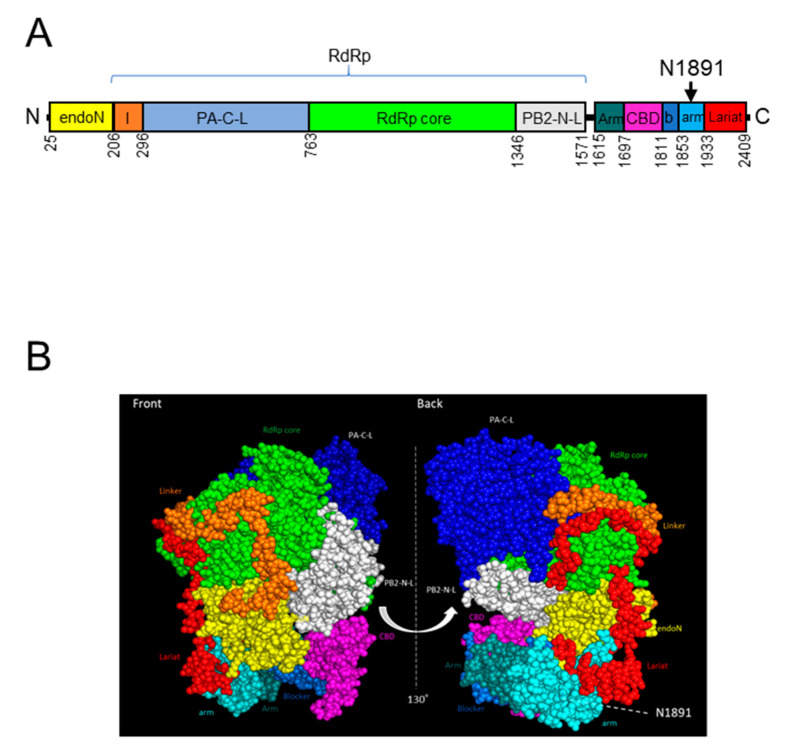
Overview of the severe fever with thrombocytopenia syndrome virus (SFTSV) L protein (**A**) The schema of the SFTSV L protein. The endonuclease domain (endo N, yellow), linker region (l, orange), PA-c-like domain (PA-C-L, blue), RdRp core domain (green), PB2-N-like domain (PB2-N-L,gray-white), Arm domain (deep teal), cap-binding domain (CBD, magenta), blocker motif (b,marine-blue), arm domain (cyan), and lariat domain (lariat, red) are shown. The position of N1891 is pointed at by the black arrow [14,15,16]. (**B**) Estimated sphere diagrams of the SFTSV L protein with patched amino acid sequences based on the reported structure model [16] are shown from two angles. Each domain is shown in the same color as A. The sidechain of position N1891 is shown in red dots, and the position of N1891 is indicated by a white dotted line.

**Figure 2 viruses-13-00033-f002:**
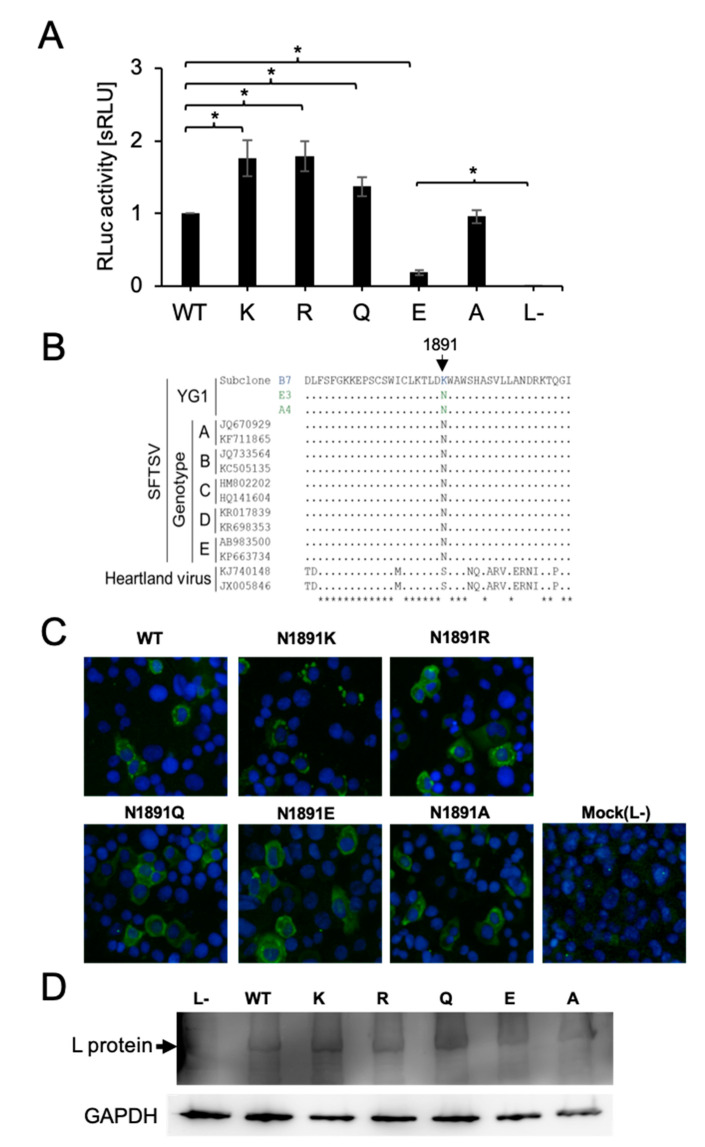
Minigenome assay for SFTSV L protein variants. (**A**) The polymerase activity of each L protein variant was measured thorough an RLuc minigenome assay in the BHK/T7-9 cells. The activities of RLuc were standardized to the FLuc activity (transfection control). Results are shown in the bar graph (the mean of the standardized relative light units [sRLU] as a fold-change value compared to WT with standard deviation in five independent transfection assays) with error bars representing the standard deviations of the means and the asterisks indicating a significant difference from 1891N (WT) or L- (negative control) by a two-tailed one sample *t*-test, as follows: * *p* < 0.01. (**B**) Amino acid sequences of SFTSV and Heartland viruses of the L protein from position 1870 to 1910 are aligned to subclone B7. Amino acids matched to subclone B7 are shown as dots. Position 1891 is indicated by the black arrow. (**C**) Indirect immunofluorescence assay for L protein variants expressed in BHK/T7-9 cells (×100). BHK/T7-9 cells were transfected with L protein variant plasmids, fixed with acetone, and stained with anti-L peptide rabbit polyclonal antibody and Alexa-488 labeled anti-rabbit IgG goat monoclonal antibody as the secondary antibody for the SFTSV L protein with DAPI for the nuclei. The merged pictures of each SFTSV L protein variant and nucleus are shown in green and cyan, respectively. (**D**) Western blotting assay for L protein variants expressed in HEK 293T cells. HEK 293T cells were transfected with L protein variant plasmids. Cell lysates were used for SDS-PAGE following a Western blotting assay with the anti-L peptide rabbit polyclonal antibody. For standardization, GAPDH was detected by the peroxidase conjugated anti-GAPDH monoclonal antibody. Black arrows indicate the bonds for the L protein.

**Figure 3 viruses-13-00033-f003:**
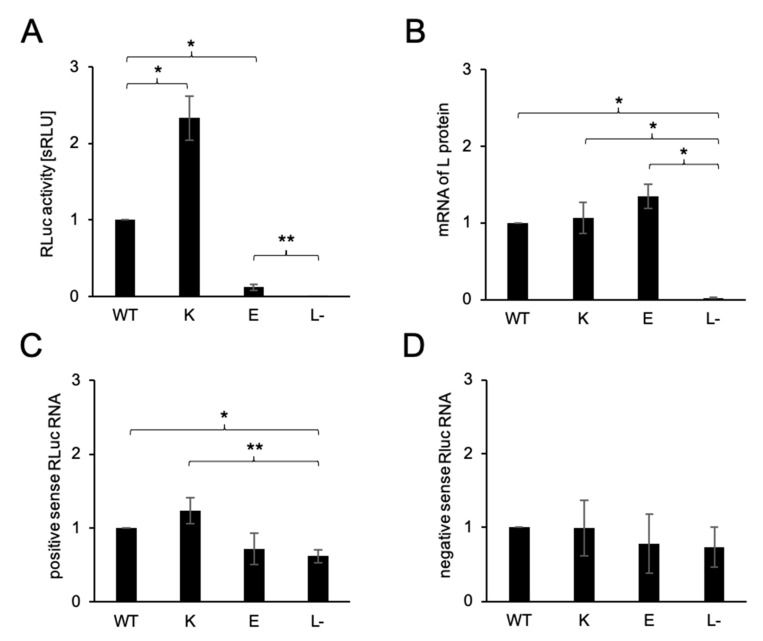
qPCR assay of RNAs under the minigenome condition (**A**) The polymerase activity of the L protein variants, WT, N1891K, N1891E, and L− (mock), was measured prior to the qPCR assay. The activities of RLuc standardized to the FLuc activity are shown in the bar graph (mean and standard deviation of the standardized relative light units [sRLU] as a fold-change value compared to WT in three independent assays) with the error bars representing the standard deviations of the means and the asterisks indicating a significant difference from WT or L− (two-tailed, one sample t-test), as follows: * *p* < 0.01, ** *p* < 0.05. The RNA samples were obtained from the same samples used for the minigenome assay. The extracted RNAs were used for RT-PCR following the qPCR assay. The amount of the L protein mRNA (**B**), positive strand Rluc RNA (**C**), and negative strand Rluc RNA (**D**) were divided by the value of FLuc to standardize the differences in transfection efficiency and cell density and are shown as the mean with standard deviation of the ratio to WT in three independent assays. Error bars represent the standard deviations of the means, and the asterisks indicate significant differences from L- (a two-tailed one sample *t*-test), as follows: * *p* < 0.01, ** *p* < 0.05.

**Figure 4 viruses-13-00033-f004:**
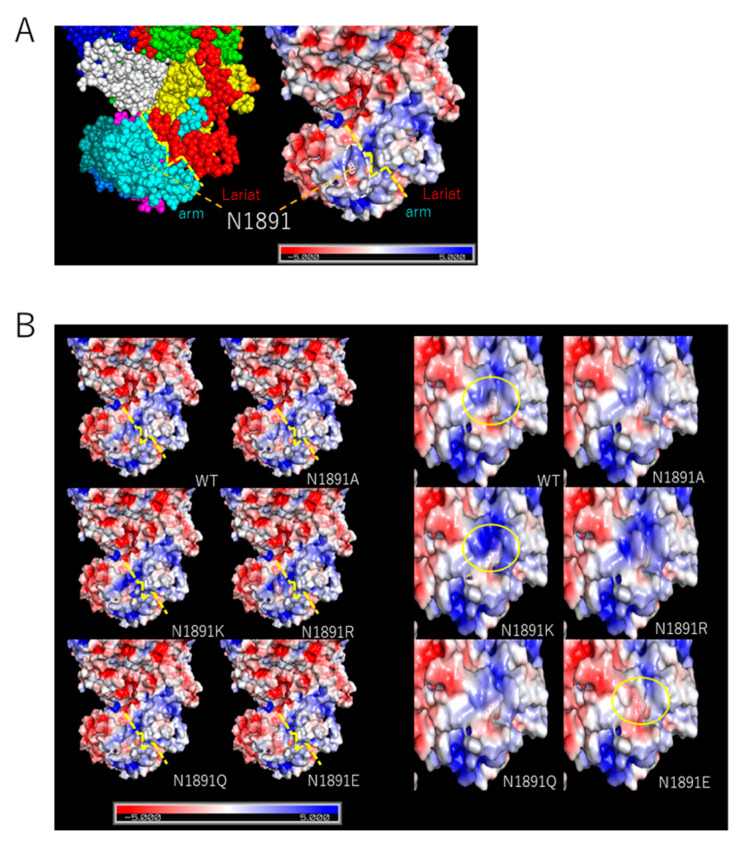
In silico analysis of the surface charges among the SFTSV L protein variants. (**A**) Sphere diagram (left) and surface charge diagram (right) of the SFTSV L protein. The deeper blue indicates a positive charge and the deeper red indicates a negative charge on the surface charge diagram. The border line of the arm domain and lariat domain is highlighted by a yellow dashed line. The position of N1891 is pointed at by the orange dashed line. The “edge” was indicated by a white dashed circle. (**B**) Comparison of the surface charge diagrams of the SFTSV L protein variants shown in the same angle as (**A**) (left) and in a different angle and zoomed (right). In both diagrams, the sidechain of position N1891 is shown in red dots. In the left diagrams, the border lines between the arm domain and the lariat domain are highlighted by wide yellow dashed lines. In the right diagrams, the surface charge differences around position 1891 were focused by a yellow circle. The deeper blue indicates a positive charge and the deeper red indicates a negative charge on the surface charge diagram.

**Table 1 viruses-13-00033-t001:** Primers used for the construction of the L protein variants.

Primer Sequence	Primer Name
AAATGGGCCTGGTCACATGCC	SFTSV LN1891K_Fw
AGGTGGGCCTGGTCACATGCC	SFTSV LN1891R_Fw
GAATGGGCCTGGTCACATGCC	SFTSV LN1891E_Fw
GCGTGGGCCTGGTCACATGCC	SFTSV LN1891A_Fw
CAATGGGCCTGGTCACATGCC	SFTSV LN1891Q_Fw
GTCAAGAGTTTTCAAGCAGATCC	SFTSV LD1890_Rv

**Table 2 viruses-13-00033-t002:** Primers used for RT-PCR and qPCR.

Primer Sequence	Primer Name
TGATCCAGAACAAAGGAAACG	RLuc 18Fw
GAAACTTCTTGGCACCTTCAAC	RLuc 820Rv
TGGGCAAATCAGGCAAATC	RLuc 242Fw
CCAAACAAGCACCCCAATC	RLuc 376Rv
AGCAGCGTCTCACCAAATCTC	SFTSV L3613Fw
GCAGGAGCTGAGCGCACTGT	SFTSV L3730Rv

## Data Availability

Data sharing is not applicable to this article.

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
