# Peer review of "The Polarity of an Amino Acid at Position 1891 of Severe Fever with Thrombocytopenia Syndrome Virus L Protein Is Critical for the Polymerase Activity"

_viruses, 2020, doi:10.3390/v13010033_

Round 1
Reviewer 1 Report
Severe fever with thrombocytopenia syndrome virus (SFTSV) is a recently recognized bunyavirus and causes a severe, often fatal, disease SFTS in humans. Currently there are no countermeasures against the disease. Noda et al. investigated a protein L of SFTSV biologically and structurally, based on their previous studies with several subclones of a SFTSV strain showing different characteristics in cell culture. The authors found that the charge status of an amino acid at position 1891 within C-terminus of L influences the protein’s activity. Because the function(s) and structure of the C-terminal domain of L has not been studied profoundly, the authors’ finding will facilitate profound understanding of L protein of SFTSV and also enhance development of countermeasures against SFTS.
The authors’ manuscript seems to be worthy to be published in the journal Viruses, but the following points should be addressed before that.
Major points
Because subclone B7 has a single mutation at position 1891 within L when compared with another subclone A4 (but not their parent) and these two subclones show different phenotypes, the author focused their study on the amino acid, Reviewer guess so. The author needs to correct description regarding this throughout the manuscript, e.g. “Compared to YG1” in Abstract.
The author described results (L294 to L319) from In Silico analysis in Discussion section. The paragraph should be moved to Results section.
Minor points
L30, please add “the” before family.
L33, one SFTS patient was reported in Taiwan, please include an article (https://wwwnc.cdc.gov/eid/article/26/7/20-0104_article) in references.
L147, please add [15] after et al.
L159, please delete a space among “whole”.
L274, please reposition the words “C-terminal region” adequately.
L274 to L276, in the sentence please specify the virus the authors are describing.
L294 to L296, the description needs reference(s).
L329 to L330, please add “might” after protein in the sentence.
L341, please put “host” before factors.
L343, please replace the word revolution with evolution.
Reviewer 2 Report
In this manuscript, the authors present an in vitro and in silico characterization of an SFTSV YG1 subclone which bears an amino acid substitution in its polymerase protein, and which showed distinct biological features in cell culture, particularly enhanced CPE effect. The manuscript provides valuable information regarding the functional and structural properties of the SFTSV viral polymerase, which still remains poorly understood, and it contributes to the global understanding of the SFTSV replication strategy.
The manuscript is well written and is clear, the methods seem appropriate and the conclusions are supported by the results and the data generated in the study.
I just have minor questions.
- There are a couple of typos in lines 71 (“open reading flame”) and 343 (“pathogenicity and viral revolution”). I believe the authors meant “open reading frame” and “pathogenicity and viral replication”.
- Materials and methods, section 2.2: the authors mention here the “inverse PCR method”. They also mention “standard molecular cloning techniques”. I believe these methods should be explained in more detail, especially the inverse PCR method, which might not be obvious for all readers. In particular, since the inverse PCR method is key to produce the variants and is not as commonly used as regular cloning procedures are, I believe the text will benefit from a more detailed description.
- Materials and methods, section 2.7: please add the appropriate reference number in line 147 for “Wang et al”.
- Also, in this section, and regarding the computational model used: the authors use a base structural model that do not include the amino acid positions under study in this work (1889 to 1896). Hence, they employ a computational algorithm to generate a new model in which these positions are deducted. Which are the limitations of this method, especially regarding the confidence in the protein model that is generated? While I don’t question the validity of this method, I believe that the text will benefit from a brief comment in the conclusions regarding the limitations of this methodology.
- Figure 1 caption: please add the appropriate reference numbers from where A and B have been generated (I believe they are 13, 14, and 15). They are correctly referenced in the text, but they should be also mentioned in the figure caption.
- Section 3.3 of the results, line 233: while it would not be necessary to evaluate all variants with the minigenome assay, I believe that the variant N1891A, which would be expected to behave in a similar trend to the WT, should be included. If available, I suggest the authors to include this information.
- Line 306 in the Discussion section: the authors mention and “edge” on the arm domain. They use this term several times in the discussion. I have found it a little hard to locate this “edge” in the figures. Is it the small red dots as shown in the supplemental figure? I suggest the authors to make a more clear definition of “edge” in the text, to facilitate its understanding.
- Line 323 in the Discussion section: the subclone B7 shows enhanced polymerase activity and clear CPE effect. Its ratio in the original YG1 virus was of 1/1,000,000 according to the authors. Do the authors have information about the fitness of the B7 subclone, specifically if the amino acid variation N1891K increases its fitness against the WT subpopulation?
- Supplemental figure: which methodology and software has been used to perform the surface charge analysis and to generate this figure? I don’t see it described in the methods section. Please add this information in the materials and methods along with appropriate references. Also add the references in the figure caption.
- The B7 subclone was selected and isolated in cell culture. I wonder about the viability of this variant in vivo. Has this variant (or a similar one with functionally equivalent substitutions) ever been observed in circulation among human SFTS cases? Do the authors have any information regarding this? I understand that YG1 was isolated form a clinical case, but I mean a virus comprised of a majoritarian N1891 variant subpopulation.
- I suggest a change in the title. Particularly the last part seems vague "... impacts on the L protein activity". I believe it should be more specific: how it does impact? This would make a more attractive title in my opinion.
